# Antimicrobial Resistance and Whole-Genome Characterisation of High-Level Ciprofloxacin-Resistant *Salmonella* Enterica Serovar Kentucky ST 198 Strains Isolated from Human in Poland

**DOI:** 10.3390/ijms22179381

**Published:** 2021-08-29

**Authors:** Tomasz Wołkowicz, Katarzyna Zacharczuk, Rafał Gierczyński, Magdalena Nowakowska, Katarzyna Piekarska

**Affiliations:** Department of Bacteriology and Biocontamination Control, National Institute of Public Health NIH—National Research Institute, 00-791 Warsaw, Poland; twolkowicz@pzh.gov.pl (T.W.); kzacharczuk@pzh.gov.pl (K.Z.); rgierczynski@pzh.gov.pl (R.G.); mrzeczkowska@pzh.gov.pl (M.N.)

**Keywords:** *Salmonella* Kentucky, antibiotic resistance, multidrug resistance, ciprofloxacin, ST198

## Abstract

Background: *Salmonella* Kentucky belongs to zoonotic serotypes that demonstrate that the high antimicrobial resistance and multidrug resistance (including fluoroquinolones) is an emerging problem. To the best of our knowledge, clinical *S.* Kentucky strains isolated in Poland remain undescribed. Methods: Eighteen clinical *S.* Kentucky strains collected in the years 2018–2019 in Poland were investigated. All the strains were tested for susceptibility to 11 antimicrobials using the disc diffusion and E-test methods. Whole genome sequences were analysed for antimicrobial resistance genes, mutations, the presence and structure of SGI1-K (Salmonella Genomic Island and the genetic relationship of the isolates. Results: Sixteen of 18 isolates (88.9%) were assigned as ST198 and were found to be high-level resistant to ampicillin (>256 mg/L) and quinolones (nalidixic acid MIC ≥ 1024 mg/L, ciprofloxacin MIC range 6–16 mg/L). All the 16 strains revealed three mutations in QRDR of GyrA and ParC. The substitutions of Ser83 → Phe and Asp87 → Tyr of the GyrA subunit and Ser80→Ile of the ParC subunit were the most common. One *S.* Kentucky isolate had *qnrS1* in addition to the QRDR mutations. Five of the ST198 strains, grouped in cluster A, had multiple resistant determinants like *bla*TEM1-B, *aac(6′)-Iaa*, *sul1* or *tetA*, mostly in SGI1 K. Seven strains, grouped in cluster B, had shorter SGI1-K with deletions of many regions and with few resistance genes detected. Conclusion: The results of this study demonstrated that a significant part of *S.* Kentucky isolates from humans in Poland belonged to ST198 and were high-level resistant to ampicillin and quinolones.

## 1. Introduction

Non-typhoidal *Salmonella* (NTS) infections are a major health problem for humans and animals worldwide. NTS are the third most frequent cause of food-borne human infections, responsible for more than 78 million illnesses and 59 thousand food-borne deaths [1]. The consumption of contaminated water and food of animal origin is considered the main source of *Salmonella* infections in humans. Travel and international trade also contributed to the global increase of *Salmonella* strains in human, food and various animal species. Although *Salmonella* infections are usually limited to uncomplicated diarrhoea, for the elderly or an immunocompromised person, they can cause severe infections, which may be life-threatening and may require an antimicrobial therapy. However, an increase in *Salmonella* strains resistant to antimicrobial agents commonly prescribed to treat salmonellosis has become a significant problem [2,3]. Fluoroquinolones (FQ) are first line drugs for treating serious *Salmonella* infections in adults. On the other hand, the emergence of FQ-resistant *Salmonella* strains has been increasingly more evident in recent years [4,5,6,7], making empiric treatment of salmonellosis difficult. Additionally, in 2017 the World Health Organization (WHO) considered FQ-resistant *Salmonella* a high-priority pathogen for the research and development of new antibiotics [8] and named FQs as the “highest priority critically important antimicrobials” for human medicine (www.who.int/foodsafety/cia/en (accessed on 15 March 2021)). Antimicrobial resistance among *Salmonella* strains is attributed to intensive and inappropriate use of antibiotics to treat both animal and human infections and may lead to resistance genes or different mobile elements (transposons, plasmids, etc.) being transferred between bacterial communities. Moreover, the application of antibiotics as growth promoters in animal feed is an important factor in the emergence of antibiotic-resistant *Salmonella* [9]. Additionally, food-producing animals, such as poultry, that are considered the main reservoir for *Salmonella* play an important role in the development and spread of antimicrobial resistance (AMR). Consequently, antimicrobial-resistant strains might be transmitted to humans via direct contact or through the food chain and the environment.

*Salmonella* enterica serovar Kentucky (*S.* Kentucky) belongs to one of the most resistant zoonotic serotypes. These bacteria are common inhabitants of the gastrointestinal systems of poultry and cows but are occasional pathogens to humans. It is worth noting that multiple antimicrobial resistance, including resistance to fluoroquinolones, is an emerging problem within this serotype. In Poland, the emergence of multi-resistant, high-level ciprofloxacin-resistant *S.* Kentucky was reported in turkeys in 2012 [10]. However, to the best of our knowledge, clinical *S.* Kentucky human strains isolated in Poland have remained undescribed to date.

In 2002, the first ciprofloxacin-resistant *S.* Kentucky strain was reported—it was isolated from a French tourist who had gastroenteritis during his stay in Egypt [11]. In recent years, highly FQ-resistant strains of *S.* Kentucky have spread worldwide [12,13]. This is largely associated with the international dissemination of a sequence type (ST198) exhibiting resistance to ciprofloxacin. *S.* Kentucky ST198 is the most frequently isolated and characterised ST associated with human infections occurring in different parts of the world [14,15,16,17].

Thus, the objective of this study was to determine the antimicrobial resistance profiles, phylogenetic relationships and multilocus sequence type (MLST) of 18 *S.* Kentucky strains collected from humans between 2018 and 2019 in Poland.

## 2. Results

### 2.1. Phenotypic Antimicrobial Resistance Profile

The MICs of 11 antimicrobials determined for the tested Salmonella Kentucky isolates are shown in Table 1. Phenotypic antimicrobial susceptibility testing indicated that among a total of 18 isolates tested, 16 isolates (88.9%) were resistant to two different antimicrobial classes: β lactams and quinolones (Table 1). The strains showed high levels of resistance to ampicillin, nalidixic acid and ciprofloxacin with MICs > 256 mg/L, ≥1024 mg/L and ≥6 mg/L, respectively (Table 1). Additionally, 7 and 6 of the 16 *S.* Kentucky isolates were resistant to tetracycline (with the MIC range 24–32 mg/L) and gentamicin (with the MIC range 8–64 mg/L), respectively. Moreover, among the 16 resistant *S.* Kentucky isolates, seven (43.7%) presented the MDR phenotype, including isolates resistant to three (*n* = 1; 6.2%) and four (*n* = 6; 37.5%) classes of antibiotics, respectively. AMP-NA-CIP and AMP-NA-CIP-TET-GEN were the most common phenotypes detected among the tested *S.* Kentucky isolates and were found in nine (56.2%) and six (37.5%) isolates, respectively. The two phenotypes mentioned were identified more frequently than AMP-NA-CIP-TET (*n* = 1; 6.25%). No resistance against cefoxitin, cefotaxime, ceftazidime, amikacin and chloramphenicol was detected (Table 2). Two *S.* Kentucky isolates (No. 85/18 87/18) had increased trimethoprim/sulfamethoxazole MIC values—1.5 mg/L and 0.75 mg/L, respectively. The compatibility between the two antimicrobial susceptibility testing methods used was observed. Interestingly, two of the 18 tested *S.* Kentucky isolates (No. 383/18 and 412/18) were susceptible to all the antimicrobial agents tested.

### 2.2. Genotypic Antimicrobial Analysis in Silico

The summary of the WGS results and antimicrobial resistance profiles is shown in Table 2. The analysis of the WGS data revealed that all the 16 antibiotic-resistant *S.* Kentucky isolates carried genes correlating with the antimicrobial resistance phenotypes determined. The majority of the tested isolates presented genes encoding resistance traits to four classes of antibiotics: quinolones, β lactams, aminoglycosides and tetracyclines. Resistance traits to sulphonamide and trimethoprim were sporadic (Table 2). No resistance genes or point mutations conferring resistance to chloramphenicol and/or colistin were found.

β-lactam resistance: All the 16 isolates resistant to ampicillin (>256 mg/L) were found to carry the blaTEM-1B β-lactamase (class A) gene. No other known molecular determinant of resistance to β-lactams was detected.

Quinolone resistance: All the 16 antibiotic resistant isolates harboured point mutations in the quinolone resistance determining region (QRDR) of *gyrA* and *parC*, conferring high-level resistance to both nalidixic acid (MICs ≥ 1024 mg/L) and ciprofloxacin (the MIC range 6–16 mg/L) of tested isolates. Point mutations involving amino acid substitutions were observed in two codons of GyrA: 83 (Ser → Phe; *n* = 16) and 87 (Asp → Tyr; *n* = 15 and Gly; *n* = 1). In ParC, substitution was also identified, but sole in codon 80 (Ser → Ile; *n* = 16). No other known quinolone resistance mutations were detected. Additionally, no substitution was found in the two quinolone-susceptible *S.* Kentucky isolates (No. 383/18 and 412/18) with the MIC of nalidixic acid amounting to 4 mg/L and of ciprofloxacin to 0.016 mg/L, respectively. Besides, isolate No. 3/19 with the ciprofloxacin MIC of 16 mg/L despite detected substitutions in GyrA (Ser83 → Phe; Asp87 → Tyr) and ParC (Ser80 → Ile) also had a plasmid mediated quinolone resistance (PMQR) determinant—qnrS1 (Table 2). No other known PMQRs genes including *qnrA*, *qnrB*, *aac(6′)-Ib-cr*, *qepA*, *oqxAB* causing low-level resistance to quinolones were detected.

Aminoglycosides resistance: Five *S.* Kentucky isolates carried the *aac(3)-Id* gene encoding resistance to gentamicin (the MICs range 8–16 mg/L). Isolate No. 3/19 with the gentamicin MIC of 64 mg/L had the *aac(3)-IId* gene that also contributes to tobramycin and netilmicin resistance. An amikacin resistance determinant—*aac(6′)-Iaa*—was identified in all the 18 *S.* Kentucky tested susceptible to amikacin (the MIC range 1.5–3 mg/L) using ECOFF or clinical breakpoint criteria. In addition, isolate No. 3/19 harboured the *aac(6′)-Iid* gene reported to confer gentamicin resistance [18]. Moreover, the WGS analysis showed the *aph(3″)-Ib*, *aph(3″)-Id* and *aadA1* genes detected in one, one, and two isolates, respectively, as only detected streptomycin resistance determinants.

Tetracycline resistance: Seven *S.* Kentucky isolates harboured the tetracycline resistance determinant *tet(A)*. It was the only tetracycline resistance gene found in the isolates tested with the tetracycline MIC of ≥24 mg/L.

Sulphonamide and trimethoprim resistance: Seven of the 16 strains of *S.* Kentucky carried genes associated with resistance to sulphonamides: *sul1* (*n* = 5) and trimethoprim: *dfrA1* (*n* = 2). Notably, two isolates with the *dfrA1* gene (No. 85/18 87/18) had the trimethoprim/sulfamethoxazole MIC value of 1.5 mg/L and 0.75 mg/L, respectively.

### 2.3. MLST, wg-SNP and wgMLST Phylogenetic Analysis

Sixteen isolates subject to WGS belonged to ST198, with MICs for ciprofloxacin amounting to ≥6 mg/L and for ampicillin >256 mg/L. The other two isolates susceptible to all the antimicrobials tested belonged to ST314 and ST696, respectively (Table 2).

The analysis of genetic similarity based on the wg-SNP analysis is presented in Figure 1 (for visibility reasons, the dendrogram is presented only for the ST198 strains). Similar results were obtained for the genotyping using wgMLST based on Enterocase database, which is why only one of these analyses is presented below (all wgMLST data, including trees in Newick format, graphics and metadata were added as a Appendix A. In the graphic file, the strains from this analysis are marked in red and labelled with the appropriate numbers). Among ST198, the strain 86/18 differed significantly (24–29 SNPs detected) from the other isolates and should be considered as an outgroup. Strains No. 145/19, 438/18 and 368/18 distinguished as cluster A but all strains in this cluster were isolated from patients living in different regions of Poland. Strains No. 188/18 and 384/19 seems to be quite similar to this cluster. On the other side of the phylogenetic tree, strains No. 85/18, 93/18, 88/18, 329/18, 169/18, 87/18, 172/18 and 325/18 are grouped into cluster B, with only 0–3 SNPs detected. Strains No. 3/19 and 457/19 are located on the separate branch.

Global comparison of wgMLST data revealed that strains grouped in mentioned cluster A could be closely related with a strain isolated in Israel in 2015 from a “wild animal” (strain named 162,835) and another strain isolated in 2015 from an unknown country (and also from unknown source, strain named 162,835_2). Strains No. 3/19 and 457/19 are located on the separate branch and are closely related with one strain isolated in 2018 in UK (named 492,521) and another two strains (but both with no specific data, named 201,403,560 and 201,403,561). Strains grouped in cluster A by wgSNP analysis were more diverse in wgMLST analysis. Strain No. 188/18 was located in a different branch and was closely related to three stains from the UK (isolated in 2015, 2018, named sam_e0b41798-f9f6-44f4-a84e-aaee07ffb6ae, 157,034 and 563,447) and one strain isolated in 2014 in Poland (named DTU2016_506_PRJ1050_Salmonella_kentucky_S14_0906). Strains No. 145/19 and 368/18 were found to be closely related with strains isolated in Czechia in 2014 (named DTU2016_461_PRJ1050_Salmonella_kentucky_S_5179) and three strains isolated in the UK in 2018 and 2020 (named 880,031, 621,471 and 1,044,964). Strain No. 438/18 was not involved in this analysis because of an error in the Enterobase.

Graphics with the phylogenetic tree based on wgMLST analysis, as well as the Newick and metadata files, are included as Appendix A.

### 2.4. SGI1-K Structure Analysis and Plasmid Detection

The *Salmonella* genomic island 1 variant K (SGI1-K) was detected in all but two isolates tested (383/18 belonged to ST314 and 412/18 belonged to ST696). None of the two distinct isolates belonged to ST198. The comparison of structure of SGI1-K from the tested isolates revealed mosaic structure and high diversity of this genomic region. The results correspond to the SNP phylogenetic analysis. The majority of the isolates had SGI1-K 33 kb in length with a lack of central region between the class 1 integron and IS26 (compared to reference SGI-K of *S.* Kentucky SRC73). These strains lacked the *tetA*, *tetR*, *aadA7*, *strA*, *strB* or mer operon. Strains No. 188/18, 368/18, 438/18, 145/19 and 384/18 had more complete SGI1-K with a length of around 47.5 kb, lacking IS1133 and IS26 regions. SGI1-K of strain No. 86/18 (length 40 kb) had no class 1 integron, mer region or Tn5393, although Tn1721 with *tetA* was present. SGI1 K in strain No. 3/19 (length 42 kb) had a similar structure, but in this case, there was also Tn5393 with the *strA* and *strB* genes.

The correspondence of the detected SGI1-K islands to the reference is shown in Figure 2.

The WGS data revealed that the detected plasmids belonged to the IncI1-I and IncR incompatibility groups and the ColRNA (Col156 and Col8282) replicon types (Table 1).

## 3. Discussion

In accordance with the EFSA/ECDC data, *S.* Kentucky is a relatively rare cause of human infections. During the last decade it was always noted in the top 10 isolates in the TESSy database, but with percentage always around 0.5% of all reported human salmonellosis cases [19]. This serotype was not noticed in Poland in years 2000–2006, but the frequency of its detection has steadily increased in recent years and this trend is reflected in epidemiological surveillance data from Poland (http://wwwold.pzh.gov.pl/oldpage/epimeld/index_p.html (accessed on 15 March 2021)). The prevalence of such isolates detected in clinical samples in Poland between 2007 and 2019 is shown in Figure 3. A large convergence of the course of both curves can be observed, especially local peaks in 2010, 2013 and 2018, which could suggest the presence of potential diffuse outbreaks of *S.* Kentucky infections in Europe in these years. Interestingly, *S.* Kentucky was the fourth most frequently detected serovar from humans in Poland in 2018 (*n* = 78) and the sixth in 2019 (*n* = 59). In order to illustrate the scale of non-typhoidal *Salmonella* cases in Poland, the total number of strains recorded in 2018 and 2019 was 9957 and 9234, respectively. However, such infections still accounted for only 0.78–0.63% of all cases of salmonellosis in Poland.

Although the first *S.* Kentucky isolate resistant to streptomycin, spectinomycin, chloramphenicol, sulfamethoxazole and tetracycline was reported in 1986 [20], to the best of our knowledge, human clinical *S.* Kentucky strains and their antimicrobial profiles remain poorly described in Poland. In this study, 18 *S.* Kentucky isolates collected in 2018 and 2019 from humans were subject to antimicrobial resistance tests. Most of the isolates (88.9%) showed high-level resistance to ampicillin (MIC > 256 mg/L) and quinolones (nalidixic acid MIC ≥ 1024 mg/L, ciprofloxacin MIC ≥ 6 mg/L) and, among them, 43.7% displayed the MDR phenotype, including isolates resistant to three and four classes of antibiotics, respectively. The spread of the quinolone-resistant *Salmonella* isolates in humans is a major health risk, especially in life-threatening infections among the elderly and immunocompromised humans that might need antimicrobial therapy. Apart from resistance to ampicillin and quinolones, resistance to tetracycline and gentamicin was also observed. Notably, according to the joint report of European Food Safety Authority (EFSA) and the European Centre for Disease Prevention and Control (ECDC), in 2018, a high percentage of human *Salmonella* Kentucky in Europe was resistant to sulphonamides (71.1%), ampicillin (72.2%) and tetracyclines (76.6%) [21]. Moreover, extremely high proportions (85.7%) of *S.* Kentucky were resistant to ciprofloxacin, with 88.6% having high-level ciprofloxacin resistance (MIC ≥ 4 mg/L) [21].

Additionally, we show that the majority of the *S.* Kentucky tested in our study belonged to ST198. ST198 is considered a high-level fluoroquinolone-resistant MDR clone that has been found in many countries of the world [11,14,15,16]. Only two of the 18 isolates tested in our study were susceptible to antimicrobial agents and belonged to other STs (314 and 696).

In Europe, human infections with MDR *S.* Kentucky ST198 were predominantly associated with travels to North Africa or Southeast Asia [12,13]. The limitation of our research was the lack of information about the travel status of patients from whom the *S.* Kentucky strains were isolated. Therefore, the original source of *S.* Kentucky ST198 tested in our study could not be determined. Additionally, our global wgMLST analysis had to be analysed very carefully because of lack of equal representativeness of the data in the Enterobase. From all 1666 *S.* Kentucky ST198 strains, 817 were isolated in the UK, 199 in the USA and 191 had no specified country of origin. Therefore, it is difficult to draw any precise conclusions that would not be an unauthorized research hypothesis. However, high-level fluoroquinolone-resistant *S.* Kentucky ST198 with the ciprofloxacin MIC of ≥ 8 mg/L was reported in Poland from turkey and pet reptiles [10]. *S.* Kentucky ST314 sensitive to antimicrobial agents has also been isolated from pet reptiles [22]. These findings together may suggest that human infections in Poland could also be acquired by domestically produced food or animal contact. The “One-Health” national surveillance and monitoring program for AMR *Salmonella* isolated from animals and humans needs to be improved to determine risk factors for the acquisition of these infections in Poland. Such an activity in identifying and monitoring of the incidence of an epidemic clone might assist in the recognition of potential *S.* Kentucky reservoirs and possibly help prevent human infections caused by drug-resistant isolates.

As mentioned above, the international ST198 isolates were identified as MDR and often have resistance determinants to ampicillin, gentamicin, tetracycline and sulfonamides, as well as high-level resistance to ciprofloxacin [12,15,17]. It should be noted that ciprofloxacin is frequently the treatment of choice for non-typhoidal salmonellosis. Resistance to this agent was originally identified in the serovars Typhimurium, Schwarzengrund and Choleraesuis and then for the first time in *S.* Kentucky isolated from a French traveller returning from Egypt in 2002 [11]. This phenotype is thought to have emerged in *S.* Kentucky ST198 in Egypt in the late 1990s [20] and has since become endemic in multiple countries, possibly in flocks of poultry. In this study, 16 of 18 human *S.* Kentucky strains were highly resistant to ciprofloxacin and belonged to ST198. All of them had substitutions detected in GyrA in codons 83 (Ser → Phe) and 87 (Asp → Tyr, Gly) and in ParC in codon 80 (Ser → Ile) and were therefore responsible for high-level ciprofloxacin resistance (Table 2). Our results are consistent with studies indicating the circulation of *S.* Kentucky isolates in Europe with similar mutations in GyrA and in ParC [12,13]. Additionally, one of the *S.* Kentucky ST198 strains tested in our study with the ciprofloxacin MIC of 16 mg/L had *qnrS1* as a sole plasmid mediating the quinolone resistance mechanism. Interestingly, *qnrS1* is the most common PMQR identified in animals or retail food in Poland [23,24]. This finding can be explained by the fact that *qnrS1* is present in the environment and humans could acquire *Salmonella* through plasmid exchange.

The second most common mechanism of resistance found in our study was TEM-1 β-lactamase, detected among all the 16 tested *S.* Kentucky ST198 clinical isolates. TEM-1 is the most commonly encountered β lactamase (penicillinase) in Gram-negative bacteria responsible for ampicillin resistance. In this study, TEM-1-producing *S.* Kentucky strains revealed high-level resistance to ampicillin (MIC > 256 mg/L). Moreover, seven of the tested *S.* Kentucky ST198 strains had the *tetA* gene that determines resistance to tetracyclines via efflux. Resistance to sulphonamides was associated with the *sul1* and *dfrA1* genes detected among five and two tested strains, respectively. All the examined strains resistant to ampicillin with high-level resistance to fluoroquinolones also had at least one aminoglycoside resistance gene. Based on our finding, the high percentage of MDR among the quinolone-resistant *S.* Kentucky isolates may be partially explained by the selective pressure and irrational/intensive use of antibiotics in livestock farming and human medicine. In this line, responsible use of antibiotics in health care and veterinary medicine remains an important pillar of AMR prevention.

Many of multidrug-resistant *S.* Kentucky ST198 observed in Europe, Africa and Asia contained the *Salmonella* genomic island 1 (SGI1) variants, particularly SGI1-K, SGI1-Q and SGI1-P. Moreover, many of these resistance determinants were found in SGI1-K, as described by Levings et al. [25]. This genomic island has a highly mosaic and diverse structure, as was shown by Hamidian et al. [26] who sequenced the most complex variant of this locus (length 52 kb). To date, there have been many described variants of SGI1-K with multiple deletions of different regions and mobile elements [27]. In our study, we found four different variants of SGI1 K. These data fully correspond to the phylogenetic analysis of these isolates. Strains from cluster A had a more complex variant of SGI1-K with a lack of only the TN5393 region, while strains from cluster B had a less complex variant of SGI1-K without the region between class 1 integron to TN5393, and a pool of resistance genes (*aadA7*, *sul1*, *tetA*, *strA*, *strB*) and the mer operon. Only one of the analysed strains (No. 3/19) had the Tn5393 region with the streptomycin resistance genes *strA* and *strB*. On the other hand, all the analysed strains had IS26 with the *bla*TEM-1 gene, in contrast to the findings by Chen et al. [27] who noticed a lack of this region in most analysed strains.

Antimicrobial resistance remains a significant problem in the world, with efforts underway in an attempt at countering the public health threat posed by bacteria resistant to antimicrobial agents. Antibiotic usage in food-producing animals has been seen as a potential source of antibiotic-resistant infections in humans, but the link between resistance in animal and human strains has not been proven. The emergence of ciprofloxacin-resistant *S.* Kentucky ST198 in Egypt has been hypothesised to be due to the common usage of fluoroquinolones in the poultry industry. The acquisition of SGI1-K encoding resistance to multiple antibiotics or plasmids harbouring antibiotic resistance determinants have resulted in increased antibiotic resistance. ST198 resistant to ciprofloxacin poses a substantial public health dilemma since fluoroquinolones are the treatment of choice for human *Salmonella* infections.

In conclusion, this study reports the first draft genome sequences of *Salmonella* Kentucky isolated from humans in Poland. The draft genome sequences provide valuable information for the antimicrobial resistance determinants, phylogenetic relationship and multilocus sequence type of the tested *S.* Kentucky isolates. Based on current results, we claim that a significant part of the clinical *S.* Kentucky tested in this study belong to the worldwide, high-level fluoroquinolone-resistant and multidrug-resistant *S.* Kentucky ST198 clone reported in many counties. The international spread of this clone is increasing and the isolation of ciprofloxacin-resistant *S.* Kentucky from food-producing animals (particularly poultry, considered the main vector) and humans will be monitored to prevent further spread of the resistant clonal strain. Further studies and the national surveillance program for AMR *Salmonella* isolates are required to determine the risk factors for the acquisition of these infections in Poland.

## 4. Materials and Methods

### 4.1. Tested S. Kentucky Isolates

Eighteen strains identified previously as *S.* Kentucky by Provincial Sanitary and Epidemiological Stations and collected at the National Institute of Public Health—National Institute of Hygiene between 2018 and 2019 were investigated. The strains were sent as part of a task in the Poland’s National Health Program for 2016–2020 “Program to confirm the correct serotypes of human *Salmonella* strains determined in Poland”. The analysed *S.* Kentucky strains were sent from different regions of Poland, as shown in Figure 4 describing their geographical distribution. One strain was isolated from urine, whereas all others were isolated from stool samples. Basic epidemiological data are presented in Table 1. All the *S.* Kentucky isolates were re-serotyped by slide-agglutination methods according to White-Kauffmann-Le Minor scheme [28] using commercial sera manufactured by Statens Serum Institut (Denmark), Biomed (Poland) and Immunolab (Poland). The confirmed strains were subject to further investigation.

### 4.2. Antimicrobial Susceptibility Testing of S. Kentucky Isolates

Antimicrobial susceptibility to 11 commonly used antimicrobials representing seven classes of antimicrobial drugs was determined by using the Kirby-Bauer disks diffusion method on the Mueller-Hinton agar medium. The following antimicrobials agents (Oxoid Ltd., Basingstoke, UK) were tested: ampicillin (10 µg), cefoxitin (30 µg), cefotaxime (5 µg), ceftazidime (10 µg), nalidixic acid (30 µg), ciprofloxacin (5 µg), gentamicin (10 µg), amikacin (30 µg), chloramphenicol (30 µg), tetracycline (30 µg), trimethoprim/sulfamethoxazole (25 µg) and interpreted according to the European Committee on Antimicrobial Susceptibility Testing (EUCAST) recommendations (http://eucast.org (accessed on 15 May 2021)). Additionally, for all the collected *S.* Kentucky strains, minimum inhibitory concentrations (MICs) for the aforementioned 11 antimicrobials were determined by using E-test strips with an antibiotic concentration gradient (bioMerieux, Marcy l’Etoile, France), with the exception of the nalidixic acid MIC determined by using the agar dilution method. The MIC results were interpreted according to the EUCAST clinical breakpoint criteria (http://eucast.org (accessed on 15 May 2021)), with the following exceptions: the clinical breakpoints for cefoxitin, nalidixic acid and tetracycline have not been determined by EUCAST, therefore only the EUCAST ECOFF values were applied (https://mic.eucast.org (accessed on 15 May 2021)). The ECOFF values allowed for classifying isolates as wild-type (WT, without phenotypically expressed resistance mechanisms) or non-wild-type (NWT, with phenotypically expressed resistance mechanisms). *E. coli* ATCC25922 control strains were included in the susceptibility test validation. Antimicrobial susceptibility for each clinical *S.* Kentucky isolate tested was measured at least twice. Antimicrobial resistance to at least three different antimicrobial classes was considered multidrug resistance (MDR).

### 4.3. Whole-Genome Sequencing Analysis

Genomic DNA for WGS purposes was isolated from an overnight culture in LB using a manual in-house procedure with the GTC, phenol extraction and precipitation steps. Isolated DNA samples were checked and quantified using a BioSpectrometer (Eppendorf, Hamburg, Germany). The libraries were constructed using the Illumina DNA Prep chemistry (Illumina, San Diego, USA) and sequenced on MiSeq (Illumina) using MiSeq Reagent Kit v3.

The raw reads were assembled using CLC Genomics Workbench 21 (Qiagen, Hilden, Germany) and using Enterobase. Except for strain No. 438/18, all other strains had good average coverage range from 34 to 112x. The *Salmonella* serotype was confirmed using SeqSero tool (CGE) [29]. MLST was assigned from the raw reads using the MLST 2.0 tool (CGE; database version 22 March 2021) [30] and SISTR1 (EnteroBase). The acquired antimicrobial resistance genes were assigned from the raw reads using ResFinder 4.1 (CGE; database from 19 February 2021) [31]. Chromosomal mutations in QRDRs were assigned manually based on the extraction of the *gyrA* and *parC* genes and the verification of the GyrA and ParC amino acid sequence alignments using CLC Genomic Workbench 21.

SGI1-K was assigned using MyDbFinder 2.0 based on the raw reads and as a database sequence of the SGI-K of *S.* Kentucky SRC73 (GenBank accession number AY463797.8) with a threshold for 60% ID and 60% of minimum length. All the sequences were extracted and aligned with the reference sequence using CLC GW 21 for further structural analysis.

The SNPs phylogenetic analysis was performed using CSI Phylogeny 1.4 (CGE) with default parameters [32]. The phylogenetic tree was drawn both for all strains and the ST198 strains. Additionally similar analysis was performed using wgMLST analysis based on EnteroBase both for only analysed strains and all 1666 *S.* Kentucky ST198 strains with WGS data deposited in the Enterobase (www.enterobase.warwick.ac.uk (accessed on 3 August 2021)). Visualisation and analysis of the generated tree was performed in Microreact (microreact.org (accessed on 4 August 2021)) [33].

## Figures and Tables

**Figure 1 ijms-22-09381-f001:**
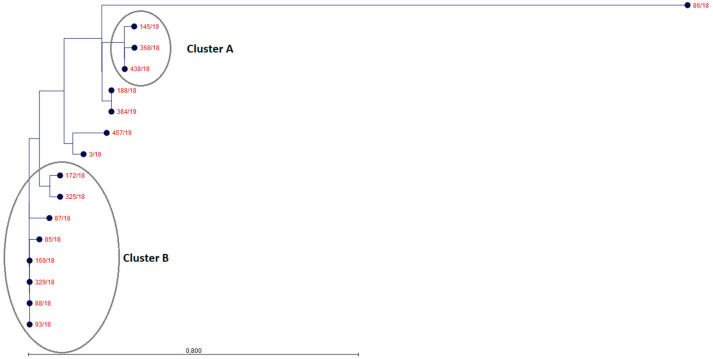
Phylogenetic tree based on the wg-SNP analysis. The dendrogram is limited only to the ST198 strains. The strain 86/18 was clearly distinct from all other isolates. Strains No. 145/19, 438/18 and 368/18 can be grouped as cluster A. Strains No. 85/18, 93/18, 88/18, 329/18, 169/18, 87/18, 172/18 and 325/18 can be grouped as cluster B.

**Figure 2 ijms-22-09381-f002:**
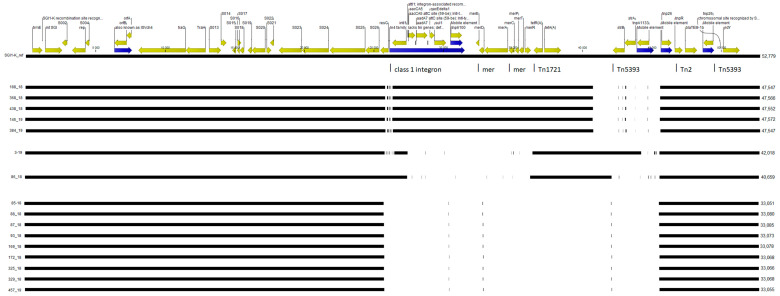
The graphical alignment of SGI1-K sequences with SGI1-K from the SRC73 strain (shown as a reference). A lack of black line means deletion of a given region of the island. Four different patterns of this locus can be distinguished, with a different pattern for mobile element deletions.

**Figure 3 ijms-22-09381-f003:**
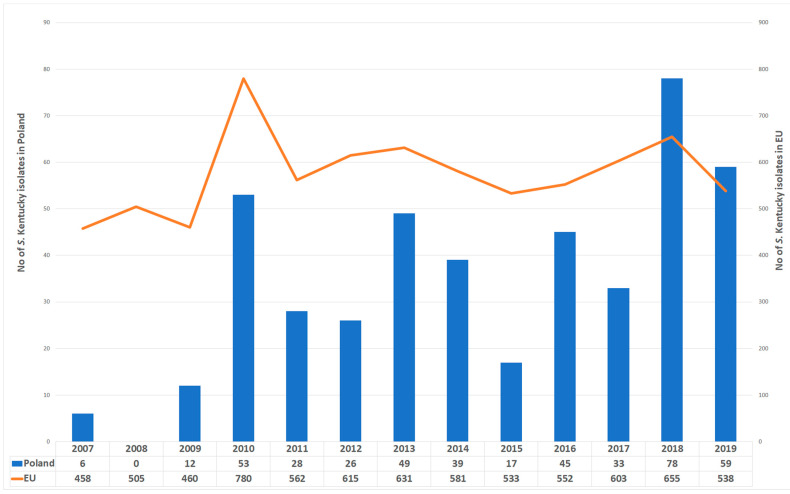
The *Salmonella* enterica serovar Kentucky isolates recovered from humans identified in Poland and in Europe, 2007–2019. These data revealed that *S.* Kentucky can be an emerging pathogen in Poland due to the constantly increasing number of recorded cases.

**Figure 4 ijms-22-09381-f004:**
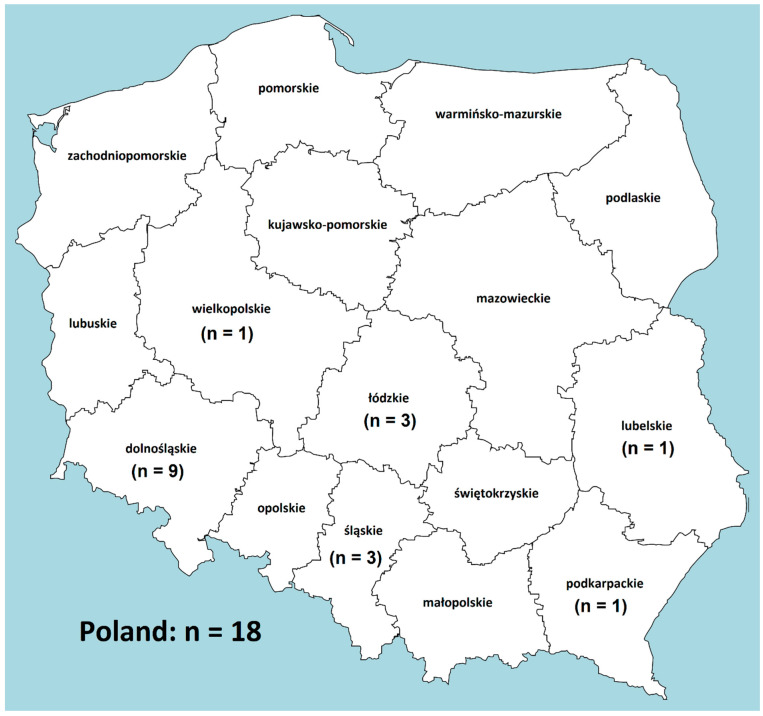
The geographical distribution of the analysed *S.* Kentucky strains isolated in 2018–2019 from patients in Poland.

**Table 1 ijms-22-09381-t001:** Basic epidemiological data and antimicrobial minimum inhibitory concentrations for the 18 *Salmonella* Kentucky strains.

No.	Isolate ID	Date of Isolation	Sex Female/Male	Age	Patient Status	Province	MIC (mg/L) ^1^										
							AMP	FOX	CTX	CAZ	GEN	AMK	NAL	CIP	TET	CHL	SXT
1	3/19	08.03.2019	M	54	patient	podkarpackie	>256	2	0.12	0.38	64	1.5	1024	16	32	2	0.047
2	169/18	ND	M	ND	patient	śląskie	>256	2	0.06	0.38	0.38	1.5	>1024	12	1.5	3	0.064
3	85/18	28.04.2018	F	12	patient	dolnośląskie	>256	2	0.06	0.38	0.38	2	1024	12	1	2	1.5
4	86/18	16.03.2018	F	35	patient	dolnośląskie	>256	2	0.06	0.38	0.38	2	1024	12	32	3	0.047
5	87/18	28.04.2018	M	75	patient	dolnośląskie	>256	3	0.12	0.25	0.38	2	>1024	12	1	2	0.75
6	325/18	09.06.2018	F	ND	patient	dolnośląskie	>256	1.5	0.06	0.38	0.25	2	>1024	12	1	3	0.064
7	329/18	15.05.2018	M	ND	patient	dolnośląskie	>256	3	0.03	0.125	0.38	2	1024	12	1	3	0.047
8	438/18	ND	M	ND	patient	wielkopolskie	>256	1.5	0.06	0.38	12	1.5	1024	12	32	2	0.19
9	188/18	05.06.2018	M	42	patient	łódzkie	>256	1.5	0.06	0.38	8	2	1024	12	32	3	0.19
10	88/18	28.04.2018	M	65	patient	dolnośląskie	>256	1.5	0.06	0.19	0.25	1.5	>1024	8	1	3	0.047
11	172/18	24.05.2018	F	ND	patient	dolnośląskie	>256	2	0.06	0.25	0.38	2	1024	8	1	2	0.064
12	384/19	July 2019	M	ND	patient	śląskie	>256	2	0.06	0.25	16	3	1024	8	32	3	0.19
13	457/19	12.09.2019	F	1	patient	łódzkie	>256	1.5	0.06	0.25	0.38	2	1024	8	0.75	2	0.047
14	93/18	27.01.2018	F	ND	carrier	dolnośląskie	>256	2	0.06	0.38	0.38	2	1024	8	1	2	0.047
15	368/18	ND	F	ND	patient	śląskie	>256	1.5	0.06	0.38	12	1.5	>1024	6	24	3	0.094
16	145/19	04.06.2019	M	ND	ND	lubelskie	>256	2	0.06	0.25	12	2	>1024	6	32	2	0.19
17	383/18	18.08.2018	F	ND	patient	dolnośląskie	0.5	2	0.06	0.19	0.25	1.5	4	0.016	1	2	0.047
18	412/18	01.10.2018	M	86	patient	łódzkie	0.38	1.5	0.03	0.19	0.25	1.5	4	0.016	1	2	0.047

ND—no data; MIC—minimum inhibitory concentration; AMP—ampicillin; FOX—cefoxitin; CTX—cefotaxime; CAZ—ceftazidime; GEN—gentamicin; AMK—amikacin; NAL—nalidixic acid; CIP—ciprofloxacin; TET—tetracycline; CHL—chloramphenicol; SXT—trimethoprim/sulphamethoxazole; ^1^ 1. MIC determination according to the EUCAST clinical breakpoint Table 2020 (mg/L)—AMP (S ≤ 8; R > 8), FOX ECOFF (S/WT ≤ 8; R/NWT > 8), CTX (S ≤ 1; R > 2), CAZ (S ≤ 1; R > 4), GEN (S ≤ 2; R > 2), AMK (S ≤ 8; R > 8), NAL ECOFF (S/WT ≤ 8; R/NWT > 8), CIP (S ≤ 0.06; R > 0.06), TET ECOFF (S/WT ≤ 8; R/NWT > 8), CHL (S ≤ 8; R > 8), SXT (S ≤ 2; R > 4); 2. MIC determination according to the EUCAST epidemiological cut-off (ECOFF) values (non-wild type >mg/L)—AMP (>8 mg/L), FOX (>8 mg/L), CTX (>0.5 mg/L), CAZ (>2 mg/L), GEN (>2 mg/L), AMK (>4 mg/L), NAL (>8 mg/L), CIP (>0.064 mg/L), TET (>8 mg/L), CHL (>16 mg/L), SXT (–).

**Table 2 ijms-22-09381-t002:** The results of phenotypic and genotypic antimicrobial resistance profile, MLST type and plasmid profile of 18 tested *S.* Kentucky isolates from humans.

No.	Isolate ID	MLST Type	Plasmids	Resistance Phenotype	Resistance Genotype by WGS								
					β-Lactams	Aminoglycosides	Sulfonamides	Trimethoprim	Tetracyclines	Fluoroquinolones			
										QRDR Amino Acid Change in			PMQR
										GyrA		ParC Ser80	
										Ser83	Asp87		
1	3/19	198	Col156 IncR	AMP, NA, CIP, TET, GEN	*bla*TEM-1B	*aac(6′)-Iaa*, *aac(6′)-Iid* *aac(3)-IId* *aph(3′’)-Ib* *aph(6)-Id*			*tet(A)*	Phe	Tyr	Ile	*qnrS1*
2	169/18	198	None detected	AMP, NA, CIP	*bla*TEM-1B	*aac(6′)-Iaa*				Phe	Tyr	Ile	
3	85/18	198	IncI1-I	AMP, NA, CIP	*bla*TEM-1B	*aac(6′)-Iaa* *aadA1*		*dfrA1*		Phe	Tyr	Ile	
4	86/18	198	Col8282	AMP, NA, CIP, TET	*bla*TEM-1B	*aac(6′)-Iaa*			*tet(A)*	Phe	Gly	Ile	
5	87/18	198	IncI1-I	AMP, NA, CIP	*bla*TEM-1B	*aac(6′)-Iaa* *aadA1*		*dfrA1*		Phe	Tyr	Ile	
6	325/18	198	IncI1-I	AMP, NA, CIP	*bla*TEM-1B	*aac(6′)-Iaa*				Phe	Tyr	Ile	
7	329/18	198	None detected	AMP, NA, CIP	*bla*TEM-1B	*aac(6′)-Iaa*				Phe	Tyr	Ile	
8	438/18	198	None detected	AMP, NA, CIP, TET, GEN	*bla*TEM-1B	*aac(6′)-Iaa* *aac(3)-Id*	*sul1*		*tet(A)*	Phe	Tyr	Ile	
9	188/18	198	None detected	AMP, NA, CIP, TET, GEN	*bla*TEM-1B	*aac(6′)-Iaa* *aac(3)-Id*	*sul1*		*tet(A)*	Phe	Tyr	Ile	
10	88/18	198	None detected	AMP, NA, CIP	*bla*TEM-1B	*aac(6′)-Iaa*				Phe	Tyr	Ile	
11	172/18	198	None detected	AMP, NA, CIP	*bla*TEM-1B	*aac(6′)-Iaa*				Phe	Tyr	Ile	
12	384/19	198	None detected	AMP, NA, CIP, TET, GEN	*bla*TEM-1B	*aac(6′)-Iaa* *aac(3)-Id*	*sul1*		*tet(A)*	Phe	Tyr	Ile	
13	457/19	198	None detected	AMP, NA, CIP	*bla*TEM-1B	*aac(6′)-Iaa*				Phe	Tyr	Ile	
14	93/18	198	None detected	AMP, NA, CIP	*bla*TEM-1B	*aac(6′)-Iaa*				Phe	Tyr	Ile	
15	368/18	198	None detected	AMP, NA, CIP, TET, GEN	*bla*TEM-1B	*aac(6′)-Iaa* *aac(3)-Id*	*sul1*		*tet(A)*	Phe	Tyr	Ile	
16	145/19	198	None detected	AMP, NA, CIP, TET, GEN	*bla*TEM-1B	*aac(6′)-Iaa* *aac(3)-Id*	*sul1*		*tet(A)*	Phe	Tyr	Ile	
17	383/18	314	None detected	-		*aac(6′)-Iaa*				WT	WT	WT	
18	412/18	696	None detected	-		*aac(6′)-Iaa*				WT	WT	WT	

AMP—ampicillin, NA—nalidixic acid, CIP—ciprofloxacin, TET—tetracycline, GEN—gentamicin, W—trimethoprim, SXT—trimethoprim/sulfamethoxazole. WGS—whole-genome sequencing; PMQR—plasmid mediated quinolone resistance; WT—wild type.

## Data Availability

All whole genome sequences were deposited in the Enterobase database (enterobase.warwick.ac.uk, accessed on 12 July 2021).

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
