# Peer review of "Antimicrobial Resistance and Whole-Genome Characterisation of High-Level Ciprofloxacin-Resistant Salmonella Enterica Serovar Kentucky ST 198 Strains Isolated from Human in Poland"

_ijms, 2021, doi:10.3390/ijms22179381_

Round 1
Reviewer 1 Report
Wołkowicz et al describe the AMR phenotypes of the S. Kentucky isolates recovered from 18 human patients in Poland during 2018-2019. The authors performed sequencing of these isolates to identify the resistance genotypes and a few additional genetic determinants including SGI-1 and plasmids. The manuscript is well written. The epidemiology of the emerging S. Kentucky ST198 strains is of interest to Salmonella researchers and researchers working in AMR. Thus, the presented work is of interest in general, however, the findings and inferences are limited in scope. Based on the MIC of fluoroquinolones, the authors conclude that the strains are likely locally acquired. While it is possible that strains are likely locally acquired, MIC data is not a strong indicator of the potential source. The scope of genomic comparison of ST198 strains isolated in this study is also limited because only the strains isolated in the study were used for comparison which does not allow one to predict the potential sources of infection. The sequences of more than 400 ST198 strains isolated globally are now available in the public database (GenBank). The authors could perform CGS analysis with publicly available strains (see the following paper: https://www.frontiersin.org/articles/10.3389/fsufs.2021.695368/abstract) for epidemiological source tracing. This will significantly enhance the impact of the current work. Moreover, the authors should consider reducing the redundancy in the results and discussion section on antibiotic resistance by removing repetitive statements or by simply combining results and discussion sections.
Line 49-58: The way these statements are written, present a direct causal relationship of antibiotic usage to resistance and implies that intensive and inappropriate antibiotic use for treatment in humans and animals leads to the development of resistance and resistance transmission. What does “intensive” and “inappropriate” use for treatment mean? It is known that the use of antibiotics for growth promotion or treatment can lead to selection for resistant populations and that there is potential for the spread of such populations of bacteria. However, I wondered, how does the use of antibiotics for treatment lead to the “development of resistance”.
Table 1 is somewhat redundant with Table 2 and mainly presents raw data for MIC so it would make more sense if table 1 is included as a supplemental file rather than in the manuscript.
Figure 1: It is unclear what method/model was used for phylogenetic analysis? The bootstrap values need to be presented at each branch of the tree. Moreover, simply comparing sequences of strains isolated from this study with each other is not epidemiologically informative. It’s unfortunate that you don’t have travel-associated information for these patients, however, you don’t necessarily need travel-associated information for patients to infer likely sources (as described in lines 241-246). It would make more sense to compare the sequences of the strains sequenced in this study with publicly available sequences of ST198 strain isolated globally (there are more than 450 strain sequences available within NCBI GenBank) to determine the relatedness of these strains with the global strain database. This will inform the sources/origins of infection in these cases.
Line 165: Please clarify how exactly the “strain 86/18 differed significantly from other isolates”?
Figure 2: It is unclear why SGI-K sequences are compared and what is gained by such comparison? It is well known that SGI-1 is a mosaic island, so gene deletion/insertion events are expected within this island.
Line 180: Please provide the ST type of the two distinct isolates (i.e., 383/18 and 412/18) here.
Line 202: If S. Kentucky was #8 serovar isolated from humans in Europe, how come it is considered a relatively rare cause of infection in Europe? Please include the average number of cases reported each year due to this serovar in Europe and in Poland for better understanding and relevance to the current study.
Line 209-210: It would make much more sense if authors calculated the percent of Kentucky relative to total NTS cases in Europe (and in Poland, if that data is available) and presented information that is easy to understand. The way these numbers are presented here, it is difficult to understand the contribution of S. Kentucky to total cases of salmonellosis caused by NTS.
Line 244-250: While it is likely that the strains could have been acquired domestically (based on MIC), however, this is not sufficient evidence of source/origin. Please revise accordingly. Again, please perform comparative genomics of strains with available strain sequences in the database as suggested above. This may provide stronger evidence of likely sources and will increase the impact of the current work.
Methods: Please clearly describe the quality criteria used for WGS analysis.
Author Response
Dear Reviewer,
Thank you for your valuable comments, which definitely helped to improve the acrylic and to catch ambiguities or errors. In fact we agree with most of them. Unfortunately, the magazine only gave us 7 days to make corrections, which, given the numerous holidays that are currently taking place, made this task extremely difficult. Some global analyzes have even started to be carried out, but getting good results would take a lot more time. However, at the same time, it seems to us that this type of analysis is the subject of an interesting global work on the epidemiology of Salmonella Kentucky.
Please find our detailed answers below Your suggestions.
Line 49-58: The way these statements are written, present a direct causal relationship of antibiotic usage to resistance and implies that intensive and inappropriate antibiotic use for treatment in humans and animals leads to the development of resistance and resistance transmission. What does “intensive” and “inappropriate” use for treatment mean? It is known that the use of antibiotics for growth promotion or treatment can lead to selection for resistant populations and that there is potential for the spread of such populations of bacteria. However, I wondered, how does the use of antibiotics for treatment lead to the “development of resistance”.
- It is also well known that every use of antibiotics generate the selective pressure on microorganisms. In our opinion, this phrase is a truism, typical and often necessary to introduce the issue at the Introduction section of the manuscript. That is why we really don’t understand the reviewer’s objections.
Table 1 is somewhat redundant with Table 2 and mainly presents raw data for MIC so it would make more sense if table 1 is included as a supplemental file rather than in the manuscript.
- Of course it can be good to include Table 1 as a supplemental file. We will ask the Editor for the opinion about that and of course we will follow the instructions.
Figure 1: It is unclear what method/model was used for phylogenetic analysis? The bootstrap values need to be presented at each branch of the tree. Moreover, simply comparing sequences of strains isolated from this study with each other is not epidemiologically informative. It’s unfortunate that you don’t have travel-associated information for these patients, however, you don’t necessarily need travel-associated information for patients to infer likely sources (as described in lines 241-246). It would make more sense to compare the sequences of the strains sequenced in this study with publicly available sequences of ST198 strain isolated globally (there are more than 450 strain sequences available within NCBI GenBank) to determine the relatedness of these strains with the global strain database. This will inform the sources/origins of infection in these cases.
- We added some clarification in the text. We regret that we do not have epidemiological data, especially on travel, to be able to link them to the presented results. Unfortunately, the very short time to correct the manuscript according to Reviewer’s suggestions prevented us from adding any broad genomics analysis. But such analysis is in fact topic for different, global publication that would analyse all the published data.
Line 165: Please clarify how exactly the “strain 86/18 differed significantly from other isolates”?
- Relevant data added in the text (both wgSNPs and wgMLST data)
Figure 2: It is unclear why SGI-K sequences are compared and what is gained by such comparison? It is well known that SGI-1 is a mosaic island, so gene deletion/insertion events are expected within this island.
- Of course it is well known that SGI-K is a mosaic island, but that does not mean that there is no point in analysing and describing it It is completely unclear for my why we shouldn’t do that.
Line 180: Please provide the ST type of the two distinct isolates (i.e., 383/18 and 412/18) here.
Relevant STs added.
Line 202: If S. Kentucky was #8 serovar isolated from humans in Europe, how come it is considered a relatively rare cause of infection in Europe? Please include the average number of cases reported each year due to this serovar in Europe and in Poland for better understanding and relevance to the current study.
- We completely agree that these sentences were inconsistent. According to Your suggestions we have added more EU data and due to that whole paragraph and Figure 3 were corrected. We believe that now it is more complex and clear.
Line 209-210: It would make much more sense if authors calculated the percent of Kentucky relative to total NTS cases in Europe (and in Poland, if that data is available) and presented information that is easy to understand. The way these numbers are presented here, it is difficult to understand the contribution of S. Kentucky to total cases of salmonellosis caused by NTS.
- As mentioned above, whole paragraph has been rewritten. Additionally we added in the text the mean percentage of S. Kentucky relative to total NTS cases in Europe (that is quite stable). We have added also some percentage information also for Polish data.
Line 244-250: While it is likely that the strains could have been acquired domestically (based on MIC), however, this is not sufficient evidence of source/origin. Please revise accordingly. Again, please perform comparative genomics of strains with available strain sequences in the database as suggested above. This may provide stronger evidence of likely sources and will increase the impact of the current work.
- We regret that we do not have epidemiological data, especially on travel, to be able to link them to the presented results. Unfortunately, the very short time to correct the manuscript according to Reviewer’s suggestions prevented us from adding any broad genomics analysis. But such analysis is in fact topic for different, global publication that would analyse all the published data.
Methods: Please clearly describe the quality criteria used for WGS analysis.
- We have add some simple clarifications
Best regards,
Reviewer 2 Report
The authors described characterization of 18 Salmonella Kentucky isolates in Poland. 16 out of 18 isolates belonged to ST198 and showed high-level resistance to ciprofloxacin with 3 point mutations in QRDR of gyrA and parC. 17 isolates had the same mutations in QRDR and were assigned in one leaf in a phylogenetic analysis where 2 clusters were identified. The reads from the ST198 isolates partly hit SGI1-K which could be responsible for multidrug resistance.
The manuscript is rather well written. Use of some technical terms would be corrected.
- “High-level resistance” would fit to fluoroquinolone resistance but not to ampicillin. It would be preferred to simply describe “resistant to ampicillin”.
- Line 17, SGI would be “Salmonella Genomic Island (SGI)”.
- Figure 1, 86/18 would be set as outgroup.
- Line 201-203, the two sentences look inconsistent. Please reconsider it.
- Lines 110, 273, “chromosomal” was not proved in this study. blaTEM-1b itself can be found in plasmids. I guess the authors used this term based on the analysis of SGI1-K. The authors should do de novo assemble and identify a contig including blaTEM-1b and a part of chromosomal gene such as those shown in the right part of Figure 2.
Author Response
Dear Reviewer,
Thank you for your valuable comments, which definitely helped to improve the acrylic and to catch ambiguities or errors. In fact we agree with most of Your suggestions and of course proper changes were made in the text. We are honoured that You have found our work interesting.
Please find our detailed answers below Your suggestions.
- “High-level resistance” would fit to fluoroquinolone resistance but not to ampicillin. It would be preferred to simply describe “resistant to ampicillin”.
- Analysed strains had MIC values for ampicillin >256 so it is in fact “high-level”. But where this amendment did not interfere much with the text, the used phrase was corrected to that proposed by the reviewer. We hope that it would be acceptable for the reviewer.
- Line 17, SGI would be “Salmonella Genomic Island (SGI)”.
- Explanation added in the text.
- Figure 1, 86/18 would be set as outgroup.
- We completely agree. Relevant sentence added in the text. Of course we can also change the Figure 1 and remove strain 86/18 to show the differences between the remaining isolates more clearly, if the Reviewer or the Editor would prefer such version of Figure 1.
- Line 201-203, the two sentences look inconsistent. Please reconsider it.
- We completely agree that these sentences were inconsistent. We have added also EU data and whole paragraph and Figure 3 were corrected. We believe that now it is more complex and clear.
- Lines 110, 273, “chromosomal” was not proved in this study. blaTEM-1b itself can be found in plasmids. I guess the authors used this term based on the analysis of SGI1-K. The authors should do de novo assemble and identify a contig including blaTEM-1b and a part of chromosomal gene such as those shown in the right part of Figure 2.
- Corrected. We agree that it would be better to identify contigs including blaTEM-1b with their wider context. But unfortunately, Illumina sequencing platform generates short reads that usually makes such analyses impossible due to repeated sequences (that usually can be found near such resistance genes). The best solution would be to perform sequencing using long-read platform (like Nanopore or PacBio) and hybrid assembly. But we haven’t got such platform in our lab and it wasn’t the main topic of the study. Maybe that would be the topic for further wider WGS analysis.
Best regards,
Round 2
Reviewer 1 Report
The authors have made minor changes to the manuscript to address few of the aesthetic issues, however the authors have not addressed my major concern/comment related to the global comparative genomics analyses to demonstrate genetic relatedness of their strains with strains available in the database. Non-availability of epi data is a limitation however it does not mean that they shouldn't perform comparative genomics analyses which should be an integral part of this study. The global comparative genomics analyses is not cumbersome or time consuming with availability of open source data and open source platforms. In the absence of this information, the study does not provide any advancement to the current knowledge of Salmonella Kentucky. I do not think that the authors justification of their inability to address a major limitation of the study due to short timeline for revision is reasonable. Speed of publication should not become hinderance to better science. Authors could easily reach out to the editor and ask for additional time if that helps them to revise the manuscript and address the major scientific limitation.
Author Response
According to reviewer’s suggestion we have added global phylogenetic analysis of public available sequences. The reviewer wrote about sequences in the GenBank database, but for genotyping of Salmonella strains it is better to use the Enterobase because this database is focus on this pathogen and such analysis. But, as we mentioned in the text, these global wgMLST data have to be analyzed very carefully because of lack of equal representativeness of the data. From all 1666 S. Kentucky ST198 strains, 817 were isolated in the UK, 199 in the USA and 191 has no specified country of origin. Unfortunately, there are also many gaps in the basic information about the samples. But such analysis and proper paragraph was added. All useful files like newick and compatible metadata file will be added as a supplement data.
Best regards,
Tomasz Wołkowicz
Round 3
Reviewer 1 Report
The authors chose the route of using wgMLST from Enterobase which is fine. They also highlight the limitations that the database is skewed. However despite these limitations they were able to provide some inferences on relatedness of their strains to the strains isolated from Israel and one from an unknown origin. While this is an improvement in in the manuscript, I suggest authors include the specific strain numbers in lines 183 (strain from Israel), 184 (strain of unknown origin), 186 (strain from UK and other two strains), 189 (three strains from UK), 190 (one strain from Poland), 191 (strain from Czechia and three strains from UK) at the least so readers know what strain from Israel and of the unknown origin they are referring to.
Moreover, authors should include at least these strains and recreate the phylogenetic tree so it is clear.
Author Response
Dear Reviewer,
Thank You for Your comment. Of course we have added these information (strains names) in mentioned paragraph. Some of these names are quite long and strange, but of course we didn’t want to interfere into them. The phylogenetic tree added as a supplementary data contains all these strains and can be easily modified or presented as a subtree, so everyone will be able to create whatever subtree they would like and would need to. We hope that it would be ok.
Best regards,
Tomasz Wołkowicz
This manuscript is a resubmission of an earlier submission. The following is a list of the peer review reports and author responses from that submission.